# Predictors of COVID-19 Stress and COVID-19 Vaccination Acceptance among Adolescents in Ghana

**DOI:** 10.3390/ijerph19137871

**Published:** 2022-06-27

**Authors:** Emma Sethina Adjaottor, Frimpong-Manso Addo, Florence Aninniwaa Ahorsu, Hsin-Pao Chen, Daniel Kwasi Ahorsu

**Affiliations:** 1Department of Behavioural Sciences, Kwame Nkrumah University of Science and Technology, Kumasi AK-4944, Ghana; eadjaottor.chs@knust.edu.gh (E.S.A.); afrimpong-manso.chs@knust.edu.gh (F.-M.A.); 2Department of Social Studies, Presbyterian University College, Akropong-Akuapem E2-0007, Ghana; florenceahorsu1@gmail.com; 3Division of Colon and Rectal Surgery, Department of Surgery, E-DA Hospital, Kaohsiung 824, Taiwan; 4School of Medicine, College of Medicine, I-Shou University, Kaohsiung 824, Taiwan; 5Department of Rehabilitation Sciences, Faculty of Health & Social Sciences, The Hong Kong Polytechnic University, 11 Yuk Choi Rd Hung Hom, Hong Kong, China

**Keywords:** fear of COVID-19, perceived stigma from COVID-19, self-stigma from COVID-19, believing COVID-19 information, preventive COVID-19 infection behaviours, COVID-19 stress, COVID-19 vaccination acceptance

## Abstract

Coronavirus disease 2019 (COVID-19) continues to ravage world economies, and with its recent mutations, countries worldwide are finding ways of ramping up their vaccination programmes. This cross-sectional design study, therefore, examined the predictors of COVID-19 stress and COVID-19 vaccination acceptance among adolescents in Ghana. A total of 817 participants were conveniently selected to respond to measures on fear of COVID-19, perceived stigma from COVID-19, self-stigma from COVID-19, believing COVID-19 information, COVID-19 infection prevention behaviours, COVID-19 stress, and COVID-19 vaccination acceptance. It was found that females believed COVID-19 information and accepted COVID-19 vaccination more than males did. Moreover, there were significant relationships between the majority of the COVID-19-related variables. Furthermore, fear of COVID-19, self-stigma from COVID-19, and COVID-19 infection prevention behaviours were found to be significant predictors of COVID-19 stress. Additionally, believing COVID-19 information, danger and contamination fears (a subscale of COVID-19 stress), and traumatic stress (a subscale of COVID-19 stress) were significant predictors of COVID-19 vaccination acceptance. These findings imply that different factors influence different COVID-19 variable. Therefore, careful considerations and research should be employed by health authorities and policymakers in preparing COVID-19 information to target different age groups and for different COVID-19 purposes.

## 1. Introduction

Coronavirus disease (COVID-19) emerged in Wuhan, China, in 2019 and has since plagued the entire world. Due to its seriousness and pervasiveness, the World Health Organisation (WHO) declared it a pandemic in 2020 [1]. The devastating effect of this pandemic on economic activities [2,3,4], health [5,6,7,8,9,10,11], and sociocultural lifestyle [12,13,14,15,16,17] around the world is very distressing. The number of COVID-19 cases keeps increasing, with the current global infection being over 513 million cases and fatalities over 6.2 million as of 5 May 2022 [18]. In Ghana, the situation is not significantly different. Ghana has over 161,000 confirmed cases with 1445 fatalities as of 2 May 2022 [19]. This has had varied inimical effects on the Ghanaian population, especially among adolescents.

Adolescence is a transitional stage from childhood to adulthood [20]. Hence, it is known to be a stage of physical and psychological turmoil where adolescents strive to fit into adult roles according to societal rules and norms. The stress faced by adolescents during this transition may, unfortunately, lead to physical and mental health challenges [21,22]. Therefore, most societies were very concerned with the physical and mental health of adolescents as most interventional programmes were halted due to the emergence of COVID-19. That is, COVID-19 preventive strategies implemented to mitigate the spread of the virus, such as lockdowns, quarantines, and physical distancing, also implied that most face-to-face contacts and actual programmes had to be cancelled [23]. This strengthened the concerns that COVID-19 may contribute to the worsening mental health conditions of adolescents [23,24,25,26,27]. Even those who were healthy before COVID-19 were reported to have increased symptoms of depression and anxiety [28,29]. Furthermore, adolescents’ perceived stress from COVID-19 was associated with mental health problems (e.g., depression and sleep disorders) and other COVID-19-related variables such as fear of COVID-19 [30,31,32]. Other studies have also examined the relationships between some COVID-19-related variables [33,34,35]. Therefore, it can be deduced that COVID-19-related variables such as fear of COVID-19, believing COVID-19 information, and COVID-19 stress may be related to each other.

During the initial phase of the COVID-19 pandemic, lockdowns, quarantines, physically distancing, and hand-washing were used as interim preventive measures to curb the spread of the virus [36,37,38]. These preventive actions, especially quarantine and physically distancing, by nature suggest the dangerousness of and fear for COVID-19 and the need to identify and withdraw from individuals infected with it. This may inadvertently taint the image and personality of such individuals even after being treated, which may further lead to stigmatisation and discrimination without proper education on COVID-19 [39,40,41]. That is, COVID-19 quarantined individuals are more likely to experience stigmatisation and discrimination, which may lead to further stress and mental health conditions [39,40,41]. Therefore, it can be deduced that adolescents may suffer in many different ways due to fake news on COVID-19, inappropriate reportage of COVID-19 issues, and the implementation of preventive measures of COVID-19 (e.g., lockdowns and quarantines). Studies on COVID-19 stigma among adolescents are inconsistent at this point [39,42,43], although it has been associated with knowledge of COVID-19 [44]. Unfortunately, these preventive measures persist as the most effective means of combating the spread of COVID-19 even after vaccinating against the virus.

The high infection and fatality rates of COVID-19 were met with accelerated development, approval, and production of its vaccines [45]. There are about 10 approved vaccines [46], with 197 vaccines at the stage of preclinical development and 154 under clinical development stages [47]. So far, 5.13 billion people have been partially vaccinated and 4.63 billion fully vaccinated (as recommended by the vaccine manufacturer) worldwide [18] as of 5 May 2022. In Ghana, there are about 9.79 million partially vaccinated and 6.12 million fully vaccinated individuals out of the 22.9 million target population [19]. With highly virulent strains of COVID-19 emerging, there has been the call to fully vaccinate all citizens, if possible. Unfortunately, in Ghana, COVID-19 vaccines were only approved for use among those aged above 15 years on 20 January 2022 [48], which covers adolescents in senior high schools (~15–17 years) but not those in junior high schools (~12–14 years). Parents, therefore, arranged for the vaccination for their wards either in the community or at school. As actual classes resumed immediately after the official announcement of vaccination of older adolescents, it is to be expected that the unvaccinated adolescents may have higher fears of contracting COVID-19 while those who contracted COVID-19 may be burdened with stigmatisation. These issues may affect their mental health and consequently affect their academic performance. Hence, school teachers and health professionals (e.g., nurses and mental health professionals) should have to monitor students’ behaviour so as to swiftly intervene to mitigate these challenges.

Researchers and healthcare professionals are also examining ways of encouraging and educating people to vaccinate themselves. Measurement scales have been developed to assess COVID-19 vaccination acceptance [49,50,51,52], which have helped researchers to objectively examine COVID-19 vaccination acceptance in various countries. Factors influencing COVID-19 vaccinations seem to be wide and varied, from external factors such as the culture, political governance, gross domestic product, and inter-country relationships [53,54,55,56,57,58,59,60,61] to internal or personal factors such as age, sex, educational levels, profession, and lifestyle behaviours [53,54,56,57,58,61,62]. For instance, previous studies have reported sex differences in COVID-19 attitudes and behaviours [63,64], leading to sex differences in COVID-19 infection [65], although other studies revealed contradictory findings on COVID-19 vaccine acceptance [61,66,67]. Due to the varied nature of the factors influencing COVID-19 vaccination acceptance, it may be necessary that countries examine the specific factors influencing their citizens’ COVID-19 vaccination acceptance as well as different age groups, such as adolescents.

This study focuses on some mental health challenges faced by adolescents due to the emergence of COVID-19 and factors that might influence their willingness to accept the COVID-19 vaccine or otherwise. Considering the paucity of findings on sex differences in some COVID-19-related variables [68], this study also examines sex differences. Specifically, this study attempts to fill in the research gaps on some COVID-19-related variables such as fear of COVID-19, perceived stigma from COVID-19, self-stigma from COVID-19, believing COVID-19 information, COVID-19 infection prevention behaviours, COVID-19 stress, and COVID-19 vaccination acceptance among adolescents in Ghana. Therefore, this study examined (i) whether there is a significant difference between males and females on COVID-19-related variables, (ii) whether there are significant relationships between these COVID-19 variables, (iii) the factors that predict COVID-19 stress among these adolescents, (iv) the factors that predict COVID-19 vaccination acceptance among these adolescents.

## 2. Materials and Methods

### 2.1. Participants and Procedure

The Kwame Nkrumah University of Science and Technology (KNUST) ethics committee (IRB ref: CHRPE/AP/203/22) approved this cross-sectional design study. The researchers conveniently selected two junior high schools and two senior high schools in the Kumasi metropolis. After the schools agreed to assist with the study, convenient dates were set for the data collection. Four different dates were agreed for the four schools. On the set date, the researchers with their research assistants (RAs) visited the classrooms of the students, informed them about the study, and provided an assent/consent form, after they were introduced to the students by a representative (e.g., assistant headmaster) of the school. All students who signed their assent/consent form were given the questionnaire package to complete. That is, guardians (e.g., housemasters/headmistresses and academic head) signed the consent forms for students below 18 years while the students signed their assent form. The students were assisted by trained RAs to handle ambiguous items during the completion of the questionnaires. Out of a total of 1100 participants who conveniently volunteered to participate in this study, 817 (74.27% response rate) completed the questionnaire. The students were not pre-informed about the study or the day of data collection. A single day was used to collect the data from each school. Fortunately, the classes visited were mostly full, especially in the senior high schools, while a few students were absent in the junior high schools. The total absentee rate was about 0.015%. All data collection procedures were executed in compliance with the Helsinki Declaration and adhered to ethical principles such as confidentiality, anonymity, and the right to withdraw without any repercussions. The participants received pens embossed with “School of Medicine, KNUST” as token of appreciation and were also thanked for participating in this study. They were also debriefed after the data collection.

### 2.2. Measures

#### 2.2.1. Demographic Characteristics

The first section of the questionnaire solicited participants’ age, sex, educational status, accommodation status, religion, and COVID-19 infection and hospitalization.

#### 2.2.2. Fear of COVID-19

The fear of COVID-19 was assessed using a seven-item Fear of COVID-19 Scale (FCV-19S). The items of FCV-19S were responded to using a five-point Likert-type scale (strongly disagree = 1 to strongly agree = 5). Its total score (summation of individual response items) ranges from 7 to 35, with higher scores indicating higher levels of fear of COVID-19. One of the sample items is “I am most afraid of corona”. It has acceptable psychometric properties [69]. The Cronbach’s alpha coefficient for this study was 0.814.

#### 2.2.3. Perceived Stigma from COVID-19

The perceived stigma from COVID-19 was assessed using an eight-item Perceived Stigma Scale from COVID-19 (PSSC). The PSSC was rated on a binary response scale (yes = 1 or no = 0). All the responses were added together to get a total score. Hence, higher scores indicate higher levels of perceived stigma. One of its sample items is “People act as if I am dishonest”. It has acceptable psychometric properties [70]. The Cronbach’s alpha coefficient for this study was 0.872.

#### 2.2.4. Self-Stigma from COVID-19

The self-stigma from COVID-19 was assessed using a nine-item Self-Stigma Scale from COVID-19 (SSSC). The SSSC was rated on a four-point Likert-type scale (strongly disagree =1 to strongly agree = 4). Hence, the responses of participants were averaged together to produce a mean score. A cut-off score of 2.5 may be used to divide participants into a high or low level of self-stigma. One of the sample items is “I feel uncomfortable because I have suspicious symptoms of COVID-19”. It has acceptable psychometric properties [70]. The Cronbach’s alpha coefficient for this study was 0.905.

#### 2.2.5. Belief in COVID-19 Information

Belief in COVID-19 information was assessed using a six-item Believing COVID-19 Information Scale (BCIS). The BCIS was rated on a five-point Likert scale (1  =  strongly disbelieve to 5  =  strongly believe). All the responses were added together to generate a total score. A higher score on the BCIS indicates a higher level of belief in the provided COVID-19 information. One of the sample items is “How much do you believe in COVID-19 information on television”. It has acceptable psychometric properties [71]. The Cronbach’s alpha coefficient for this study was 0.780.

#### 2.2.6. COVID-19 Infection Prevention Behaviours

The COVID-19 infection prevention behaviours were assessed using a five-item Preventive COVID-19 Infection Behaviours Scale (PCIBS). The PCIBS was developed based on the preventive behaviours recommended by the World Health Organization Q&A on coronaviruses (COVID-19) [72]. That is, WHO advised individuals to engage in these behaviours to avoid COVID-19 infection. These infection prevention behaviours are made up of five items, which were responded to on a five-point Likert scale (1 = almost never to 5 = almost always). These responses were added together to generate a total score. A higher score on the PCIBS indicates performing preventive behaviours more frequently. One of the sample items is “How often do you stay home when you feel unwell”. It has acceptable psychometric properties [71]. The Cronbach’s alpha coefficient for this study was 0.707.

#### 2.2.7. COVID-19 Stress

The COVID-19 stress was assessed using a 36-item COVID-19 Stress Scale (CSS). It basically measures COVID-19-related distress, which is made up of five subscales including COVID-19-related danger and contamination fears (12 items), fears of socio-economic consequences (six items), xenophobia (six items), traumatic stress symptoms about COVID-19 (six items), and compulsive checking and reassurance seeking (six items). The CSS was rated on a five-point Likert scale (0 = not at all/never to 4 = extremely/almost always). Participants’ responses were averaged to get an overall mean score. Higher mean scores indicate higher levels of COVID-19-related distress, as well as for the respective subscales. It has acceptable psychometric properties [73]. One of the sample items is “I am worried about catching the virus”. The Cronbach’s alpha coefficient for this study was 0.946, and 0.848 to 0.886 for the subscales.

#### 2.2.8. COVID-19 Vaccination Acceptance

The COVID-19 vaccination acceptance was assessed using 12-item Motors of COVID-19 Vaccination Acceptance (MoVac-COVID19). The items of the MoVac-COVID19 were rated on a 7-point Likert scale response format (strongly disagree = 1 to strongly agree = 7). The responses were summed up to get a total score, with a higher score indicating a higher level of COVID-19 vaccine acceptance. The MoVac-COVID19 has acceptable psychometric properties [49,50,51,52]. One of the sample items is “It is important that I get the COVID-19 jab”. The Cronbach’s alpha coefficient for this study was 0.650. 

### 2.3. Data Analysis

The demographic information of participants was presented using descriptive statistics such as means and standard deviations (M ± SD), and frequencies with their respective percentages. An independent *t* test was also used to test the significant difference between males and females on the various COVID-19 variables. Furthermore, relationships between all COVID-19 variables were examined using Pearson *r*. In addition, two hierarchical linear regression models were used to examine the factors that predict COVID-19 stress and COVID-19 vaccine acceptance among adolescents in Ghana after controlling for demographic characteristics (i.e., age, sex, educational level, accommodation status, and religion). The statistical analyses were conducted using SPSS version 23 software for Microsoft Windows (Armonk, NY, USA: IBM Corp). The level of significance was set at 0.05. 

## 3. Results

The participants (n = 817) of this study had a mean age of 16.10 (SD = 1.68) years, with the majority being males (54.7%), in senior high school (61.1%), in a boarding house accommodation (53.6%), and Christians (92.4%). In addition, only seven (0.9%) of the participants had had a COVID-19 infection, and four (0.5%) were hospitalized because of COVID-19 infection (see Table 1).

Table 2 shows differences between males and females on the various COVID-19 variables. The findings indicate that females (M ± SD, 20.78 ± 4.53) have significantly higher scores on believing COVID-19 information (*t*(815) = 2.312, *p* = 0.021) than those of their male counterparts (19.91 ± 5.88). Moreover, females (56.38 ± 12.55) have significantly higher scores on COVID-19 vaccination acceptance (*t*(815) = 2.447, *p* = 0.015) than those of their male counterparts (54.38 ± 10.77). Even though there is no significant between-group difference on the general COVID-19 stress scores (*t*(815) = 0.012, *p* = 0.991), at the subscale level, males (1.04 ± 0.97 and 1.59 ± 1.09, respectively) have significantly higher scores on traumatic stress (*t*(815) = −4.012, *p* < 0.001) and compulsive checking (*t*(815) = −1.964, *p* = 0.050) compared to their female counterparts (0.78 ± 0.83 and 1.44 ± 1.01, respectively). All the other between-group comparisons are not significant.

Table 3 shows the relationships between all of the COVID-19-related variables used in this study. The findings indicate that there are significant positive relationships (*r* = 0.098–0.912, *p* < 0.05) between all the variables except between (i) perceived stigma from COVID-19 and believing COVID-19 information, and preventive COVID-19 infection behaviours, (ii) believing COVID-19 information and traumatic stress (subscale of COVID-19 stress), and (iii) COVID-19 vaccination acceptance and fear of COVID-19, perceived stigma from COVID-19, self-stigma from COVID-19, preventive COVID-19 infection behaviours, COVID-19 stress, socio-economic consequences (subscale of COVID-19 stress), xenophobia (subscale of COVID-19 stress), contamination (subscale of COVID-19 stress), and compulsive checking (subscale of COVID-19 stress). Moreover, there is a significant negative relationship between COVID-19 vaccination acceptance and traumatic stress (*r* = −0.090, *p* = 0.01).

Table 4 shows the predictive factors of COVID-19 stress and COVID-19 vaccination acceptance of the participants. Factors that significantly predict COVID-19 stress are fear of COVID-19 (standardised coefficient (*β*) = 0.324, *p* < 0.001), self-stigma from COVID-19 (*β* = 0.200, *p* < 0.001), and preventive COVID-19 infection behaviours (*β* = 0.102, *p* = 0.002), which account for 23% of all possible predictive factors for COVID-19 stress (*F*(12, 767) = 19.055, *p* < 0.001). For COVID-19 vaccination acceptance, believing COVID-19 information (*β* = 0.125, *p* = 0.001), danger and contamination fears (a subscale of COVID-19 stress, *β* = 0.140, *p* = 0.025), and traumatic stress (a subscale of COVID-19 stress, *β* = −0.176, *p* < 0.001) are its predictive factors, which account for 6.1% of all possible predictive factors (*F*(17, 762) = 2.910, *p* < 0.001). All the other factors are not significant.

## 4. Discussion

The present study focused on the factors associated with predicting COVID-19 stress and COVID-19 vaccine acceptance among adolescents specifically in Ghana. Therefore, it sought to examine (i) whether there is a significant difference between males and females on COVID-19 variables used in this study, (ii) whether there are significant relationships between these COVID-19 variables, (iii) the factors that predict COVID-19 stress among these adolescents, and (iv) the factors that predict COVID-19 vaccination acceptance among these adolescents. 

The findings revealed that females believed information on COVID-19 and accepted COVID-19 vaccination more than males did. Hence, it can be suggested that females may be more willing to accept information on COVID-19 and subsequently vaccinate against the virus. The study further suggests that males and females have different ways and/or levels of believing COVID-19 information, and so health professionals should take note of the difference in order to boost COVID-19 vaccination acceptance among both sexes. On the other hand, at the COVID-19 stress subscale levels, the study revealed that males had higher levels of traumatic stress and compulsive checking compared to their female counterparts, although there was no significant difference between the groups on the overall COVID-19 stress. This indicates that males are more likely to have nightmares, intrusive thoughts, or emotion-laden images related to COVID-19 due to either direct or vicarious traumatic exposure to COVID-19 and using maladaptive responses such as compulsive checking and reassurance seeking [73], which may exacerbate their COVID-19 stress levels. The current findings are consistent with previous studies that reported sex differences in COVID-19 attitudes and behaviours [63,64], leading to sex differences in COVID-19 infection [65], although other studies revealed contradictory findings on COVID-19 vaccine acceptance [61,66]. In sum, this study affirms that there are sex differences and that female adolescents are more likely to perceive COVID-19 as a dangerous health problem and be more prepared to comply with the available scientific preventive protocols, which may enhance the effectiveness of dealing with the pandemic.

There were significant relationships between the COVID-19-related variables used in this study with small to large effect sizes [74,75]. For instance, fear of COVID-19 related positively with all the other COVID-19 variables studied except COVID-19 vaccination acceptance. This indicates that as the levels of fear of COVID-19 increase, the levels of the other COVID-19 variables may increase and vice versa. Other important significant relationships existed as well, although there were a few that were not significant; for example, there was no significant relationship between perceived stigma from COVID-19 and believing in COVID-19 information, and COVID-19 infection prevention behaviours. Moreover, COVID-19 vaccination acceptance did not relate to fear of COVID-19, perceived stigma from COVID-19, self-stigma from COVID-19, COVID-19 infection prevention behaviours, and COVID-19 stress. These suggest that not all COVID-19-related variables directly relate to each other, especially in regard to COVID-19 vaccination. Further studies are needed to examine the factors that mediate the association between other COVID-19-related variables and COVID-19 vaccine acceptance. These findings are not entirely new as some studies have also examined and established similar relationships between some COVID-19-related variables [32,33,34,35,44].

Further analysis revealed that fear of COVID-19, self-stigma from COVID-19, and COVID-19 infection prevention behaviours were the significant predictors of COVID-19 stress. In other words, increased levels of fear of COVID-19, self-stigma from COVID-19, and COVID-19 infection prevention behaviours may lead to higher COVID-19 stress levels among adolescents in Ghana. In as much as some level of fear of COVID-19 is needed to help with mitigating the pandemic, higher levels of fear of COVID-19 may be detrimental to the fight against the COVID-19 pandemic as they may lead to COVID-19-related distress among the adolescents. In addition, higher levels of self-stigma from COVID-19 may impair self-motivation, which may lead to extra stress on individuals to comply with COVID-19 infection prevention behaviours protocols. It is also noteworthy that stress is capable of further triggering other health conditions [22]. Therefore, health professionals should take into consideration all these factors when educating citizens on COVID-19 and its preventive measures. Parents and teachers may assist in properly informing adolescents or students on issues of COVID-19 and mitigating their COVID-19 stress. To the best of the researchers’ knowledge, this is the first study to examine COVID-19-related variables that influence COVID-19 stress among adolescents. A study among healthy adults and people with schizophrenia revealed a significant positive association between fear of COVID-19 and COVID-19 stress [35]. A closely related study that examined predictive factors of post-traumatic stress disorder (PTSD) revealed that stigma was not able to predict PTSD, which is contrary to this study’s finding [70].

In addition, believing in COVID-19 information, danger and contamination fears, and traumatic stress (both being subscales of COVID-19 stress) were significant predictors of COVID-19 vaccination acceptance. It is reassuring to know that information on COVID-19 influences one’s acceptance of COVID-19 vaccination. That is, good, appropriate, and credible information on COVID-19 may influence people to accept the COVID-19 vaccine. Previous studies have also emphasized the usefulness of accurate information and education for COVID-19 vaccination [34,56,58]. Among countries in the West African region, it was revealed that knowing the effectiveness and safety of COVID-19 vaccines increased the willingness to vaccinate among adults [67]. Moreover, danger and contamination fears as a subscale of COVID-19 stress influenced adolescents’ willingness to accept the COVID-19 vaccine. This shows that appropriate levels of danger and contamination fears may be imbued into information on COVID-19 vaccination by health professionals and educators to help influence adolescents’ acceptance of vaccination. In fact, a previous study revealed that danger and contamination concerning COVID-19, irrespective of the severity (i.e., mild, moderate, and severe), positively influenced COVID-19 vaccination intent [62]. Moreover, traumatic stress, another subscale of COVID-19 stress, was found to negatively predict COVID-19 vaccination acceptance. That is, intrusive thoughts or nightmares of the COVID-19 pandemic due to the direct and/or vicarious exposure to disturbing images from media sources may negatively influence adolescents’ willingness to accept the COVID-19 vaccine. This is consistent with a previous study that revealed that negative thoughts about the COVID-19 vaccine negatively influence vaccination [34]. Therefore, extreme caution should be exercised in showing distressing and gruesome images on COVID-19 to the general public and especially adolescents. In extreme cases where distressing and gruesome images are to be shown, viewers should be cautioned ahead of the presentation with emotionally sensitive images masked as much as possible. 

### Limitations

This study has some limitations. A cross-sectional design was used for this study, which implies that only associations among variables can be inferred and not causation. Therefore, readers and health professionals should take note of this and apply the findings appropriately. Longitudinal studies may also be conducted to strengthen these findings. Furthermore, the participants used are adolescents and so the findings may not reflect the views of the adult population. Future studies may examine adults on these variables so as to make the findings generalizable for all age groups of the population. In addition, self-report measures were used to gather the data and this may be prone to social desirability bias. Nevertheless, the measures put in place during data collection (e.g., adequate information on the study, confidentiality, and anonymity) and the robust psychometric properties of the measures suggest that the data have an appreciable degree of trustworthiness.

## 5. Conclusions

The present cross-sectional study revealed that females believed information on COVID-19 and accepted COVID-19 vaccination more than males did. Males, on the other hand, had higher levels of traumatic stress and compulsive checking compared to their female counterparts. Moreover, there were significant relationships between the majority of the COVID-19-related variables. Furthermore, fear of COVID-19, self-stigma from COVID-19, and COVID-19 infection prevention behaviours were found to be the significant predictors of COVID-19 stress. In addition, believing in COVID-19 information, danger and contamination fears, and traumatic stress (both being subscales of COVID-19 stress) were the significant predictors of COVID-19 vaccination acceptance. These findings imply that different factors influence different COVID-19 variable. Therefore, careful considerations and research should be employed by clinicians, health communicators, researchers, and policymakers in preparing COVID-19 information so as to target different age groups and for different COVID-19 purposes. Finally, parents and guardians have greater roles to play in guiding their adolescents in making appropriate COVID-19 vaccination decisions and in mitigating COVID-19 stress.

## Figures and Tables

**Table 1 ijerph-19-07871-t001:** Participants’ demographic characteristics (n = 817).

	Mean ± SD or n (%)	Missing n
Age (years)	16.10 ± 1.68	17
Sex		
Females	370 (45.3%)	
Males	447 (54.7%)	
Education		
Junior High	318 (38.9%)	
Senior High	499 (61.1%)	
Accommodation		21
Day	358 (43.8%)	
Boarding	438 (53.6%)	
Religion		12
Christian	755 (92.4%)	
Moslems	44 (5.4%)	
Traditional	5 (0.6%)	
Others	1 (0.1%)	
COVID-19 infection		179
Yes	7 (0.9%)	
COVID-19 infection hospitalization		179
Yes	4 (0.5%)	

**Table 2 ijerph-19-07871-t002:** Sex differences on the COVID-19-related variables.

Variables	Females	Males	*t*	df	*p*
	M ± SD	M ± SD			
Fear of COVID-19	20.17 ± 6.06	19.95 ± 6.38	0.489	815	0.625
Perceived stigma from COVID-19	3.46 ± 2.87	3.19 ± 2.77	1.359	815	0.174
Self-stigma from COVID-19	2.20 ± 0.80	2.15 ± 0.76	1.079	815	0.281
Believing COVID-19 information	20.78 ± 4.53	19.91 ± 5.88	2.312	815	0.021
Preventive COVID-19 infection behaviours	16.46 ± 3.78	16.34 ± 4.68	0.390	815	0.697
COVID-19 stress	1.53 ± 0.80	1.53 ± 0.85	0.012	815	0.991
Danger and contamination	1.70 ± 0.92	1.63 ± 0.97	1.011	815	0.312
Socio-economic consequences	1.75 ± 1.21	1.60 ± 1.12	1.799	815	0.072
Xenophobia	1.79 ± 1.13	1.66 ± 1.08	1.601	815	0.110
Traumatic stress	0.78 ± 0.83	1.04 ± 0.97	−4.012	815	<0.001
Compulsive checking	1.44 ± 1.01	1.59 ± 1.09	−1.964	815	0.050
COVID-19 vaccination acceptance	56.38 ± 12.55	54.38 ± 10.77	2.447	815	0.015

**Table 3 ijerph-19-07871-t003:** Correlation matrix of the studied variables.

		1	2	3	4	5	6	6a	6b	6c	6d	6e	7
1	Fear of COVID-19	-											
2	Perceived stigma from COVID-19	0.144 **	-										
3	Self-stigma from COVID-19	0.286 **	0.451 **	-									
4	Believing COVID-19 information	0.276 **	0.059	0.184 **	-								
5	Preventive COVID-19 infection behaviours	0.253 **	0.037	0.119 **	0.251 **	-							
6	COVID-19 stress	0.416 **	0.183 **	0.330 **	0.150 **	0.206 **	-						
6a	Danger and contamination	0.368 **	0.200 **	0.312 **	0.130 **	0.153 **	0.912 **	-					
6b	Socio-economic consequences	0.255 **	0.127 **	0.222 **	0.112 **	0.107 **	0.795 **	0.673 **	-				
6c	Xenophobia	0.320 **	0.137 **	0.278 **	0.121 **	0.130 **	0.835 **	0.754 **	0.639 **	-			
6d	Traumatic stress	0.378 **	0.098 **	0.218 **	0.045	0.196 **	0.671 **	0.512 **	0.367 **	0.393 **	-		
6e	Compulsive checking	0.342 **	0.127 **	0.257 **	0.181 **	0.265 **	0.706 **	0.497 **	0.422 **	0.459 **	0.534 **	-	
7	COVID-19 vaccination acceptance	0.027	−0.008	0.033	0.145 **	0.032	0.029	0.060	0.021	0.047	−0.090 *	0.036	-
	Mean	20.05	3.31	2.17	20.30	16.40	1.53	1.66	1.67	1.72	0.92	1.52	55.29
	SD	6.24	2.82	0.78	5.336	4.30	0.82	0.95	1.17	1.11	0.92	1.06	11.64

* *p* < 0.05, ** *p* < 0.01.

**Table 4 ijerph-19-07871-t004:** Factors predicting adolescents’ COVID-19 stress and COVID-19 vaccination acceptance.

	COVID-19 Stress				COVID-19 Vaccination Acceptance			
	B	SE	*B*	*p*-Value	B	SE	*B*	*p*-Value
Step 1								
(Constant)	1.764	0.458		<0.001	59.927	6.441		<0.001
Sex (males)	0.006	0.060	0.003	0.923	−2.068	0.838	−0.089	0.014
Age	−0.018	0.029	−0.037	0.543	−0.202	0.414	−0.029	0.626
Education (Senior high)	0.090	0.147	0.054	0.538	1.019	2.060	0.043	0.621
Accommodation (Day)	−0.027	0.127	−0.016	0.833	−1.272	1.783	−0.055	0.476
Religion								
Islamic	0.136	0.129	0.038	0.294	−3.582	1.819	−0.070	0.049
Traditional	−0.692	0.370	−0.067	0.062	1.666	5.202	0.011	0.749
Others	−0.103	0.824	−0.005	0.900	−2.761	11.584	−0.009	0.812
Step 2								
(Constant)	0.021	0.438		0.961	48.236	6.909		<0.001
Fear of COVID-19	0.043	0.005	0.324	<0.001	−0.022	0.079	−0.012	0.777
Perceived stigma from COVID-19	0.013	0.010	0.044	0.214	−0.229	0.164	−0.055	0.165
Self-stigma from COVID-19	0.211	0.039	0.200	<0.001	0.403	0.623	0.027	0.517
Believing COVID-19 information	−0.001	0.005	−0.008	0.805	0.274	0.084	0.125	0.001
Preventive COVID-19 infection behaviours	0.020	0.006	0.102	0.002	0.029	0.103	0.011	0.775
COVID-19 stress								
Danger and contamination	-	-	-	-	1.714	0.763	0.140	0.025
Socio-economic consequences	-	-	-	-	−0.430	0.497	−0.043	0.387
Xenophobia	-	-	-	-	−0.017	0.602	−0.002	0.978
Traumatic stress	-	-	-	-	−2.229	0.596	−0.176	<0.001
Compulsive checking	-	-	-	-	0.687	0.510	0.063	0.178
R^2^ (Adjusted R^2^)	23% (21.8%)				6.1% (4%)			
ΔR^2^	22.2%				4.3%			
Δ*F*	44.113 ***				3.496 ***			

Sex, age, educational level, accommodation, and religion were adjusted in the models. *** *p* < 0.001.

## Data Availability

Data are available on reasonable request from the corresponding author.

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
