# Peer review of "Predictors of COVID-19 Stress and COVID-19 Vaccination Acceptance among Adolescents in Ghana"

_ijerph, 2022, doi:10.3390/ijerph19137871_

Round 1

Reviewer 1 Report

The paper is well written, and the intention is good. However, sections of the paper need to be more focused.

The introduction leading to the statement of the objectives was too broad. A suggestion is to focus on adolescent behaviour and perceived factors affecting adolescent mental health and behaviour since the start of the pandemic. 

For example, how did the information about the availability or approval of10 vaccines (worldwide??) and the 100+ more vaccines in the vaccine manufacturing pipeline affect adolescents' mental health? 

 What type of information were the adolescents reacting to? It is difficult to know from the paper whether world news about COVID-19 preventive measures of lockdowns, quarantines, physically distancing, and hand-washing constitute the COVID-19 information being investigated or whether it was local preventive practices, local media images, world wide web social images. A clearer scope is needed since the paper is directed at policy making to boost vaccination.

The paper should adopt a working definition for full vaccination and should be consistent in discussing the statistics. The discussion of Ghana vaccination figures includes booster dose (does that include first and second boosters as done in some other countries). The world statistic discussed did not mention any booster dose. It is good to stay consistent. 

Access to vaccines was staggered (opened to different age groups at different times in different countries while access to different types of vaccines also vary) so it will be good to state when the participants in the study had access to vaccines in Ghana. 

The paper should clarify the following.

Whether the survey took place on the single set date, were students informed prior to the day, what was the absentee rate on the day of survey, and if those absent were more likely to be COVID-19 positive or isolating due to close contact with COVID case. 

Whether headmaster(s) was the guardian for students below 18 years -possibly those boarding. 

 The mean age of participants was 16.10+/-1.68.  Intuitively, the younger participant (below 18 years of age) could describe or evaluate their level of stress and fear but would they be making decisions about accepting vaccinations? Were school guardians (headmasters involved in decisions to vaccinate below 18 years old? Please provide clarification.

It will be good to include some of the items or indicators of the measures used under the subsection Measures and discuss some of the indicators as done for demographic characteristics and stress. They will make the paper easier to read and to follow by the audience.

 Overall, the paper is well written. It could be easier to read and followed if the above suggestions are considered.  

Author Response

We thank the reviewer for the insightful comments. We have revised the manuscript appropriately by providing a point-by-point response to the reviewer’s comments in blue font colour.

Reviewer 2 Report

General practice is to have articles described in literature review and then relate findings to literature review. New information is discussed for the first time in the discussion, with no reference to literature review. 

Background information about number vaccinated in Ghana indicates number vaccinated, but doesn't compare to overall population or segments of population vaccinated. No comparison to other countries in the region. Research doesn't study the impact of family / government policy on stress and attitude toward vaccination. 

Author Response

(The authors gave the same response as above.)
